# Achieving Equiaxed Transition and Excellent Mechanical Properties in a Novel Near-β Titanium Alloy by Regulating the Volume Energy Density of Selective Laser Melting

**DOI:** 10.3390/ma17112631

**Published:** 2024-05-29

**Authors:** Xiaofei Li, Huanhuan Cheng, Chengcheng Shi, Rui Liu, Ruyue Wang, Chuan Yang

**Affiliations:** 1School of Mechanical Engineering, Shandong University of Technology, Zibo 255000, China; lixiaofei20210629@163.com (X.L.); 19862512385@163.com (R.L.); wangruyue2024@163.com (R.W.); 2Luoyang Ship Material Research Institute, Luoyang 471023, China; chenghuanhuan@725.com.cn; 3Institute of Machinery Manufacturing Technology, China Academy of Engineering Physics, Mianyang 621900, China

**Keywords:** selective laser melting, near-β titanium alloy, columnar-to-equiaxed transition, precipitated phase, tensile property

## Abstract

This research investigated the relationship between volume energy density and the microstructure, density, and mechanical properties of the Ti-5Al-5Mo-3V-1Cr-1Fe alloy fabricated via the SLM process. The results indicate that an increase in volume energy density can promote a transition from a columnar to an equiaxed grain structure and suppress the anisotropy of mechanical properties. Specifically, at a volume energy density of 83.33 J/mm^3^, the average aspect ratio of β grains reached 0.77, accompanied by the formation of numerous nano-precipitated phases. Furthermore, the relative density of the alloy initially increased and then decreased as the volume energy density increased. At a volume energy density of 83.33 J/mm^3^, the relative density reached 99.6%. It is noteworthy that an increase in volume energy density increases the β grain size. Consequently, with a volume energy density of 83.33 J/mm^3^, the alloy exhibited an average grain size of 63.92 μm, demonstrating optimal performance with a yield strength of 1003.06 MPa and an elongation of 18.16%. This is mainly attributable to the fact that an increase in volume energy density enhances thermal convection within the molten pool, leading to alterations in molten pool morphology and a reduction in temperature gradients within the alloy. The reduction in temperature gradients promotes equiaxed grain transformation and grain refinement by increasing constitutive supercooling at the leading edge of the solid–liquid interface. The evolution of molten pool morphology mainly inhibits columnar grain growth and refines grain by changing the grain growth direction. This study provided a straightforward method for inhibiting anisotropy and enhancing mechanical properties.

## 1. Introduction

Titanium alloys have been widely utilized in the aerospace industry due to their exceptional combination of low density, high strength, and good fracture toughness [1,2]. Among them, near-β titanium alloy is commonly employed in large load-bearing components such as fuselage docking frames and landing gear beams due to its ability to achieve ultra-high strength [3,4]. The selective laser melting (SLM) process is a typical 3D printing technology for layer-by-layer printing that can overcome the limitations of traditional manufacturing processes to achieve near-net forming of lightweight and complex structures [5,6,7]. Therefore, the high-performance near-β titanium alloy produced via the SLM process is of great significance for realizing the double-weight reduction of aerospace vehicle materials and structures.

However, the steep temperature gradient and extremely high cooling rate of SLM lead to the formation of coarse columnar grains, resulting in significant anisotropy and the degradation of mechanical properties in the alloy. Therefore, achieving the controllable preparation of fine equiaxed microstructures through SLM is crucial for improving performance and ensuring isotropy behavior. Currently, several strategies have been employed to control the grain structure during SLM. The first strategy involves modifying alloy composition, with a focus on eutectic elements such as Fe and Cu that have significant grain refinement effects [8]. The second strategy aims to inhibit the grain growth during SLM by adding a second phase to increase heterogeneous nucleation points; examples include graphene [9] and nano-BN powder [10]. The third strategy involves parameter optimization [11,12]. In principle, the modification of alloy composition and parameter optimization is primarily based on the interdependence theory [13,14]. This theory suggests that the increased constitutive supercooling in front of the solid–liquid interface promotes nucleation and is beneficial for the equiaxed transition and refinement of columnar grains. Constitutive supercooling mainly promotes the formation of equiaxed grains by increasing the nucleation rate at the leading edge of the grain growth interface. Previous studies have demonstrated that introducing strong β stable elements with a high growth limiting factor [15] (*Q* = *m_l_c*_0_(*k* − 1)) as solutes in titanium alloys can eliminate β columnar grains, refine grains, and improve the comprehensive mechanical properties [16]. Furthermore, the refinement of grain size is highly influenced by the solute content in the alloy. Excessive solute content can result in the formation of numerous brittle precipitates, leading to a significant reduction in the ductility of the material [17]. The addition of a second phase primarily aims to promote equiaxed grain transformation by increasing heterogeneous nucleated particles. Among these, ceramic particles that react in situ to produce TiB whiskers play a dominant role. Han, C. J. et al. [18] incorporated insoluble B_4_C ceramic particles into titanium alloys to provide heterogeneous nucleated particles, thereby promoting the formation of equiaxed grains. However, whether through alloying or second-phase addition, there are limitations in significantly enhancing material strength while ensuring a certain level of plasticity. High alloying and a high content of the second phase will promote the formation of brittle phase in the alloy and seriously reduce the plasticity of the alloy.

Based on the solidification theory, the grain morphology of the β phase during the SLM process is primarily influenced by the thermal gradient (*G*) and solidification rate (*V*) [19,20]. Since the thermal environment is dynamic, equiaxed grain formation can only be achieved when *G* is sufficiently reduced to allow constitutive supercooling to occur [21]. Additionally, it has been observed that low *G*/*V* values are more likely to facilitate the transition from columnar to equiaxed grains, in conjunction with the relationship between solidification growth rate and the thermal gradient [22]. It should be noted that printing parameters during the printing process greatly affect the *G*/*V* value [23,24]. Studies have indicated that volume energy density encompasses four key printing parameters: laser power (*P*), scanning speed (*v*), scanning layer thickness (*t*), and scanning spacing (*h*) [25]. Therefore, the temperature gradient can be directly reduced by adjusting the volume energy density, and the transformation from coarse columnar grains to equiaxed fine grains of nearly-β titanium alloys can be promoted.

At the same time, it is important to note that the printing parameters significantly impact the porosity of SLM alloys. High porosity can greatly reduce the plasticity of the material and limit its application in SLM. Additionally, during the SLM process, the ω transition phase is precipitated in the near-β titanium alloy. The ω phase serves as an auxiliary nucleation point for the α phase, leading to a significant increase in the strength of the near-β alloy.

In this study, the near-β titanium alloy was chosen as the research focus. Printed samples were prepared using selective laser melting (SLM) under different printing parameters to investigate the influence of volume energy density on microstructure and mechanical properties. The study also aimed to uncover the mechanism by which volume energy density affects the transformation of columnar grains to equiaxed grains and the evolution law of nucleation of ω phase-assisted precipitation phase. The findings of this study can serve as a foundation for the preparation of high-performance near-β titanium alloys using SLM; to realize the application of lightweight, high-strength, isotropic near-β titanium alloy in the aerospace field; and to promote the development of structural materials in this field.

## 2. Experiment

### 2.1. Experimental Material and Equipment

The powder feedstock used in this study was prepared using the plasma rotating electrode method and was provided by the Northwest Nonferrous Metals Research Institute, referred to as Ti-55311 in this paper. The specific chemical composition of the powder is presented in Table 1. The SEM image of the powder and the particle size distribution are depicted in Figure 1. The powder surface was smooth with good sphericity and no significant irregular or hollow particles were observed. Most of the powder particle size was in the range of 15~53 μm, meeting the requirements for the SLM process.

The experiment was conducted using an SLM 125 HL printer prepared by Nikon SLM Solutions AG (Lübeck, Germany). A schematic diagram of the SLM processing is depicted in Figure 2a. The device was equipped with an IPG fiber laser transmitter with a wavelength of 1070 nm and a spot diameter of 70 μm. The building chamber had dimensions of 125 × 125 × 125 mm^3^. Throughout the SLM process, the scanning strategy and layer thickness (30 μm) remained constant, and the substrate preheating temperature was maintained at 150 °C. Figure 2b illustrates the scanning strategy, with the laser beam rotation angle set at 15° to minimize residual stresses in the fabricated specimens. In this study, the twenty-five sets of process parameters for laser power (220~300 W), scanning speed (900~1300 mm/s), and scanning spacing (60~140 μm) were obtained through orthogonal experiments. Additionally, five-volume energy density parameters—40.29 J/mm^3^, 64.93 J/mm^3^, 83.33 J/mm^3^, 111.11 J/mm^3^, and 155.56 J/mm^3^—were selected to investigate the evolution of microstructure as expressed by [26]:*E* = *P*/*vht*(1) Here, *E* represents the volume energy density (J/mm^3^), *P* denotes the laser power (W), *v* stands for the laser scanning speed (mm/s), *h* indicates the scanning spacings (μm), and *t* represents the layer thickness (μm). In Figure 2c, a cubic specimen with dimensions of 10 × 10 × 10 mm^3^ is depicted. Additionally, Figure 2d,e illustrate the specific dimensions of the tensile specimens.

### 2.2. Characterization

All printed samples were wire-cut, ground, and polished separately. Phase analysis of samples under different volume energy density conditions was performed using X-ray diffraction. X-ray diffraction (XRD, Bruker D8 Focus, Bruker AXS, Karlsruhe, Germany) is performed using D, with Max-RB instruments with Cu Kα radiation. The step size was set to 2°/min, and the scanning angle range was 20~90°. After polishing, the density of the sample was characterized using an optical microscope. Subsequently, the surface ofsample was corroded using Kroll reagent (5 mL HF + 25 mL HNO_3_ + 50 mL H_2_O) for 15~30 s, and the sample microstructure was observed after ultrasonic cleaning for 5 min. Optical microscope (OM, XD30M, SOPTOP, Ningbo, China) and field emission scanning electron microscope (SEM, Quanta 250FEG, FEI, Hillsboro, USA) were used to observe the metallographic structure and fracture morphology. Field emission transmission electron microscopy (TEM, Tecnai G2F 20, FEI, Hillsboro, USA) and high-resolution transmission electron microscopy (HRTEM) were used to further characterize the microstructure and morphology of the alloy. The crystallographic characterization of the matrix phase was conducted through electron backscatter diffraction (EBSD), the data acquired on Nordlys-Ⅱ& Channel 5.0 System equipment manufactured by HKL Technology Co., Ltd. from Oxford, EU. The scanning step size was 0.2 μm and an operating voltage was 20 KV. OIM Analysis TM V8.2 software was used to analyze the crystal information of the sample matrix phase. The specimens were prepared through electrolytic polishing and ionized in a liquid nitrogen environment for about 1 min. The room-temperature tensile properties were measured using a universal testing machine with a tensile strain rate of 1 × 10^−3^ s^−1^. At the same time, in order to enhance the accuracy of the data, three sets of repeatability experiments were conducted for each parameter.

## 3. Results

### 3.1. Columnar-to-Equiaxed Transition

Figure 3 illustrates the grain structures in the as-printed Ti-55311 alloy samples produced under different volume energy density conditions. It is evident that volume energy density plays an important role in the morphological evolution of β grains, leading to the transition from columnar to equiaxed grains. As shown in Figure 3a, at low volume energy density conditions, the microstructure comprises a layered arrangement with one layer of oblique β columnar grains and another layer of equiaxed grains. The formation of oblique β columnar grains is attributed to the predominant heat flow direction along the deposition direction and is inclined towards the laser scanning direction [22]. In addition, at low volume energy density conditions, the content of the equiaxed grain is extremely low. Conversely, samples prepared under high volume energy density conditions predominantly exhibit a near-equiaxed grain structure. The aspect ratio of the alloy is less than 0.5 under low volume energy density but can reach approximately 0.8 under high volume energy density, indicating a near-equiaxed grain microstructure, as shown in Figure 4f. Furthermore, as shown in Figure 4b, there is a significant increase in the width of the columnar grains at 64.93 J/mm^3^, which is consistent with the previous reports on SLM-processed titanium alloys [27]. However, the epitaxial growth of grains is inhibited at this stage. At 83.33 J/mm^3^, the coarse columnar grains disappear and transform into fine needle-like β grains growing in the building direction, as shown in Figure 4c. When the volume energy density reaches 111.11 J/mm^3^, the aspect ratio of the grain reaches its peak, which is about 0.87. However, with a further increase in the volume energy density, there is a decrease in aspect ratio and the epitaxial growth of the grains occurs, as shown in Figure 4e.

Figure 5 illustrates the size distribution of β grains at different volume energy densities. The Gaussian fitting results indicate that as the volume energy density increases, the grain size distribution of the alloy undergoes a transition from large grains to small grains, and then back to large grains, as depicted in Figure 5a. Similarly, the average grain size in Figure 5b follows this trend. With the volume energy density increased from 40.29 J/mm^3^ to 83.33 J/mm^3^, the average grain size of the alloy decreases significantly, reaching 63.92 μm. Combined with Figure 4b, it can be inferred that the reduction in grain size is attributed to the transformation of alloy grains from columnar grains to equiaxed grains. The grain size experiences a notable increase within the volume energy range of 111.11 J/mm^3^ to 155.56 J/mm^3^, even reaching 103.9 μm at a volume energy density of 155.56 J/mm^3^. This indicates that high volume energy density can promote grain growth by enhancing both the setting time and energy of the molting pool.

Figure 6 shows the EBSD results of SLM-processed samples prepared at various volume energy densities. It is evident from the results that the alloy exhibits a significantly low texture strength at high volume energy densities, attributed to equiaxed grain transition, as shown in Figure 6c–e. In contrast, the specimens prepared at low volume energy densities display a distinct texture along the <001> direction, as depicted in Figure 6a,b.

Figure 7 presents grain boundary maps for Ti-55311 alloy at different volume energy densities. As depicted in the figure, the grain consists mainly of high-angle and low-angle grain boundaries. With an increase in volume energy density, there is an initial rise followed by a decrease in the number of low-angle grain boundaries. The higher heat input leads to significant internal stress within the alloy, resulting in an increased presence of low-angle grain boundaries within the alloy grains. It is worth noting that a higher volume energy density and more sufficient solidification time lead to the formation of more stress-free β grains in SLM-processed Ti-55311 alloy, thereby resulting in fewer or even no low-angle grain boundaries being distributed in β grains [4]. Low-angle grain boundaries primarily exist within the grain, effectively impeding dislocation movement and improving the strength of the alloy [28].

### 3.2. Precipitated Phase

To determine the phase composition of the Ti-55311 alloy fabricated via SLM with different volume energy densities, XRD measurements were conducted on the sample surface. As depicted in Figure 8, the β phase with a body-centered cubic lattice structure was detected clearly in all samples. It is noteworthy that a small amount of α/α′ phases precipitated under low volume energy density and medium volume energy density conditions.

The relative intensities of the precipitated phases in XRD are extremely low and could not be directly observed via SEM. Therefore, TEM was utilized to observe the precipitated phases at different volume energy densities, as shown in Figure 9. In the yellow circle of Figure (a, e, and h), the V-shaped configuration of α precipitates was observed, and this configuration exhibits the lowest energy and is referred to as a self-accommodation mechanism that can significantly enhance mechanical properties [29,30]. At 40.29 J/mm^3^, the amount and size of the precipitate phase were smaller, as shown in Figure 9a–c. When the volume energy density increased to 83.33 J/mm^3^, there was a significant increase in both the content and size of the precipitated phase, as depicted in Figure 9d–f. However, with a further increase in volume energy density, the content of the precipitated phase suddenly decreased, while its size increased to the micron level, as shown in Figure 9g–i. Aluminum (Al) is an α stable element with an atomic radius smaller than that of other β stable elements. When volume energy density increases, Al is more likely to be dissolved to inhibit the precipitation of α phases. Furthermore, an increase in volumetric energy density provides energy for grain growth and promotes needle-like α grain growth. The in-depth TEM examination of samples further revealed phase types and an orientation relationship between the β matrix phase and precipitated phase. The TEM SAD pattern showed that the orientation relationship between the β phase and α phase was [−111]_β_//[010]_α_. It should be noted that the ω phase precipitated in the β matrix at a volume energy density of 83.33 J/mm^3^. The diameters of the ω precipitates ranged from 20~50 nm and were comparable to the general sizes of the isothermal ω (10~20 nm) but larger than those of the athermal ω (<5 nm). This indicates that the ω precipitates in the current sample should be classified as isothermal ω phase [31]. The ω phase is considered an unstable phase and is formed by the lattice collapse of the (110) crystal plane of the β phase. It is important to note that, as shown in Figure 9f, the (100) plane of the ω phase is fully coherent with the (002) plane of the α phase, suggesting that it provides nucleation particles for the α phase and assists its nucleation [32]. However, at higher or lower volume energy densities, there is no presence of an unstable ω phase in this alloy, which may be attributed to local thermal conditions.

### 3.3. Relative Density

In addition to microstructure, density also significantly influences the comprehensive performance of alloys. Figure 10 illustrates the porosity distribution and statistical plots of Ti-55311 alloy under different volume energy densities, highlighting the laser volumetric energy density represented by the microstructures in red font. At both higher and lower volume energy densities, the alloy exhibits low density, and yet, the overall relative density remains above 99%, as shown in Figure 10j. This suggests that the present alloy possesses a wide SLM processing window. When the volume energy density is low, irregular holes are observed in the alloy, as shown in Figure 10a. These irregular lack-of-fusion (LOF) holes at low volume energy density result from incomplete powder melting inside the alloy and are typically located between tracks and layers in multi-orbital metal printing processes such as SLM [33]. As the volume energy density increases, these pores transition from irregular LOF to nearly spherical tiny pores. However, excessive volume energy density can lead to metal vapor volatilizing and producing a large number of holes [34], as demonstrated in Figure 10i.

### 3.4. Mechanical Properties

In order to investigate the impact of the precipitated phase and matrix phase on mechanical properties, tensile tests were conducted at 40.29 J/mm^3^, 83.33 J/mm^3^, and 155.56 J/mm^3^. Additionally, the influence of equiaxed grain transition on the anisotropy was examined by comparing mechanical properties along and perpendicular to the building direction. As shown in Figure 11, all samples showed high yield strength (YS > 900 MPa). Specific strength and plasticity values under different conditions are presented in Table 2. Notably, samples fabricated perpendicular to the building direction at 83.33 J/mm^3^ demonstrated an exceptionally high yield strength (~1003.06 MPa) and good elongation (~18.16%). However, at a volume energy density of 155.56 J/mm^3^, alloy strength and plasticity were significantly reduced, leading to obvious intergranular fracture behavior. It is worth mentioning that there was little difference in the mechanical properties along and perpendicular to the building direction under conditions of 83.33 J/mm^3^ and 155.56 J/mm^3^ energy densities. These results indicate that the increasing volume energy density was beneficial to eliminate the anisotropy of the sample induced by SLM.

Figure 12 depicts the fracture morphology of Ti-55311 alloy at various volume energy densities. At low volume energy densities, cracks and cellular dendrites are observed, severely reducing the elongation of the alloy during the tensile process at room temperature, as shown in Figure 12a,b. As the volume energy density increases, cellular dendrites disappear and a large number of dimples appear, indicating ductile fracture mode, as depicted in Figure 12c,d. This ductile tearing behavior can consume more energy, effectively inhibiting the crack propagation and improving ductility [35]. However, when the volume energy density reaches 155.56 J/mm^3^, both the number and size of dimples in the alloy decrease significantly, with large cleavage steps becoming predominant. This suggests a decrease in plasticity for the alloy.

## 4. Discussion

### 4.1. On the Mechanism of Columnar-to-Equiaxed Transition

The present experimental results demonstrate that the β phase, precipitated phase, and density of Ti-55311 alloy are greatly affected by the volume energy density during the SLM process. The volume energy density affects the development of β grains’ structure by influencing the size, morphology, and thermal distribution of the melt pool. The results indicate that both the length and depth of the melt pool increase linearly with an increase in volume energy density, with a greater change rate in the depth compared to length [36]. As shown in Figure 13j, the length-to-depth ratio of the melt pool gradually decreases with the increasing volume energy density and tends to stabilize. According to the solidification theory, the microstructure during SLM is mainly affected by the temperature gradient (*G*) and the solidification rate (*V*). Bermingham, M. et al. [21] demonstrated that a decrease in *G* and an increase in *V* promote the transition from columnar to equiaxed grains. Under the condition of low volume energy density, a “disc-shaped” molten pool is formed with large *G* value, promoting columnar grain formation as depicted in Figure 13a,b. As the volume energy density increases, the melt pool morphology changes to “deep cup-shaped”. The large melt pool inhibits the formation of columnar grains by increasing heat loss during the SLM process, resulting in a fan-shaped grain structure [31], as shown in Figure 13c,d. Shi, R. P. et al. [37] found that the temperature gradient at the center line was much lower than that at the fusion line inside the melt pool. As shown in Figure 13e–h, it is observed that under the condition where the volume energy density is higher than 83.33 J/mm^3^, the grain morphology in the melt pool is composed of equiaxed grains. The mathematical modeling by Knapp, G. L. et al. [38] also showed that an increase in volume energy density can reduce the local temperature gradient in the melt pool and promote the formation of equiaxed grains. At the same time, the higher volume energy density will also lead to a higher temperature in the pre-deposited layers and pre-melt channels, which will promote the epitaxial growth of grains between multiple channels or multi-layered melt pools and increase the grain size, as depicted in Figure 13i. Based on the Kurz–Fisher model [39], the columnar spacing *λ* can be calculated using the following equation:(2)$$\lambda =BV^{-1/4}G^{-1/2}=BT^{-1/4}G^{-1/4}$$ Here, *B* represents the material-dependent parameter, *V* denotes the solidification rate, *G* indicates the temperature gradient, and T stands for the cooling rate. The grain size is inversely proportional to the solidification rate, temperature gradient, and cooling rate. It is observed that the columnar spacing of columnar β grains increases with the rise in volume energy density, as illustrated in Figure 13a,b.

In addition to this, the higher volume energy density can also reduce the *G* by controlling the flow of liquid in the melt pool. The metal liquid creates the thermocapillary flow in the molting pool with low to high surface tension, known as Marangoni convection. Antony and Arivazhagan [40] simulated the Marangoni effect under different volume energy density conditions, demonstrating that an increase in volume energy density enhances the Marangoni effect. This effect leads to increased liquid flow in the melt pool, continuously removing the heat accumulated at the solidification front of the melt pool, thereby reducing the temperature gradient and promoting the nucleation of equiaxial grains [41].

As shown in Figure 14a,b, the formation of equiaxed β grains is attributed to the increase in volume energy density, leading to a reduction in temperature gradient and an increase in constitutive supercooling at the solid–liquid interface. When the volume energy density is low (40.29 J/mm^3^), the alloy consists of tiny columnar β grains and a small amount of nano α precipitated phases. With a medium volume energy density (83.33 J/mm^3^), the alloy is characterized by fine equiaxed β grains and numerous nano α phases. At high volume energy density (155.56 J/mm^3^), coarse equiaxed β grains dominate with only a few micron α phases present, as shown in Figure 14c. Additionally, TEM analysis reveals an unstable ω phase at medium volume energy density, which facilitates the nucleation of the α phase.

Overall, these findings shed light on how variations in volume energy density impact grain structure and phase composition within the alloy system.

### 4.2. Strengthening Mechanisms

The current experimental results demonstrate that varying grain structures can lead to different mechanical properties at different volume energy densities. To clarify the contribution of distinct strengthening mechanisms to the yield strength of the alloy, the following equation can be employed for evaluation:(3)$$\sigma_{YS}=\sigma_{0}+\sigma_{GB}+\sigma_{p}$$ Here, *σ*_0_ represents the critical resolved shear stress of pure Ti (=180 MPa), *σ_GB_* denotes the grain boundary strengthening, and *σ_P_* signifies the precipitate strengthening.

The *σ_GB_* is usually evaluated by the classical Hall–Petch equation:(4)$$\sigma_{GB}=k\cdot d^{-\frac{1}{2}}$$ Here, *k* is the grain boundary strengthening coefficient and *d* is the average grain size. The *d* of the present sample is derived from the EBSD analysis. The columnar-equiaxed-transition of β grains can refine the grains, resulting in a reduction in the grain size from 105.258 μm to 63.92 μm. This results in a 1.3-fold increase in the fine-grained strengthening of the alloy.

According to Li et al. [42], a significant number of nano-precipitated phases can hinder the movement of dislocations; facilitate interaction, interlocking and accumulation between dislocations; and improve the strength and plasticity of alloys. The strength contribution from the nano-precipitated phase can be estimated as below [43,44,45]:(5)$$\sigma_{p}=\frac{0.13G_{m}b}{\gamma }\ln\frac{r}{b}$$
(6)$$\lambda \approx d_{p}\left[{\left(\frac{1}{2V_{p}}\right)}^{\frac{1}{3}}-1\right]$$ Here, *G_m_* and *b* represent the shear modulus and Burgers vector of the matrix, respectively; *r* is the radius of α precipitates (*r = d_p_*/2); *γ* is the anti-phase boundary energy of α precipitates; *V_p_* is used to indicate the volume fraction of reinforcement nano-precipitates; and *λ* represents the interparticle spacing. It can be inferred from this that increasing the volume fractions and decreasing the sizes of nano-precipitates will lead to an improvement in the yield strength of the Ti-55311 alloy.

Therefore, at a condition of 83.33 J/mm^3^, the alloy exhibits a yield strength of 1003.06 MPa and an elongation of 18.16%. The rapid cooling rate during this process results in the formation of numerous low-angle grain boundaries within the grain structure, which impedes the dislocation movement and enhances the comprehensive mechanical properties of the alloy. Under the condition of 155.56 J/mm^3^, the alloy displays an equiaxed microstructure but with a reduced presence of low-angle grain boundaries and significantly increased grain size. This leads to brittle fracture behavior and a notable decrease in the comprehensive mechanical properties of the alloy.

The in-plane anisotropy (*IPA*) value was used to quantitatively depict the anisotropy in mechanical properties as shown below [46]:(7)$$IPA=\frac{\left(T_{H}-T_{V}\right)}{\frac{\left(T_{H}+T_{V}\right)}{2}}\times 100$$ Here, *T_H_* and *T_V_* represent the mechanical properties along the building direction and perpendicular to the building direction. If the *IPA* value is equal to 0, then the mechanical properties are isotropic. As the *IPA* value increases, the anisotropic is enhanced. After calculation, it was found that at volume energy density values of 40.29 J/mm^3^, 83.33 J/mm^3,^ and 155.56 J/mm^3^, the *IPA* values were 5.9, 0.74, and 3.07, respectively. This indicates that equiaxed grains can significantly reduce the anisotropy of alloys.

## 5. Conclusions

The titanium alloys manufactured via SLM usually show the microstructures of coarse columnar grains and unevenly distributed phases, resulting in low and anisotropic mechanical properties, which are not conducive to application in the aerospace field. By regulating the volume energy density, the pore defects and microstructure of the material can be controlled, the properties of the alloy can be improved, and the isotropy of the alloy can be achieved. The following conclusions are drawn:High energy density can promote the transition from columnar to equiaxed grains by decreasing the length-to-depth ratio of the melt pool, reducing the temperature gradient, and increasing the Marangoni effect.In the SLM process, different precipitated phases were formed at different volume energy densities, which had different strengthening effects on the Ti-55311 alloy. It is worth noting that at the volume energy density of 83.33 J/mm^3^, numerous ω phases precipitated, providing a large number of nucleated particles for the α phase.At the volume energy density of 83.33 J/mm^3^, the Ti-55311 alloy achieved extremely high strength and plasticity, with a yield strength of 1003.06 MPa, and an elongation of 18.16%.

## Figures and Tables

**Figure 1 materials-17-02631-f001:**
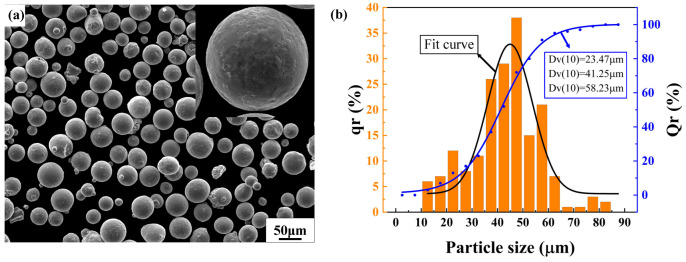
The powder feedstock: (**a**) SEM image for atomized powder particles; (**b**) the distribution of particle size.

**Figure 2 materials-17-02631-f002:**
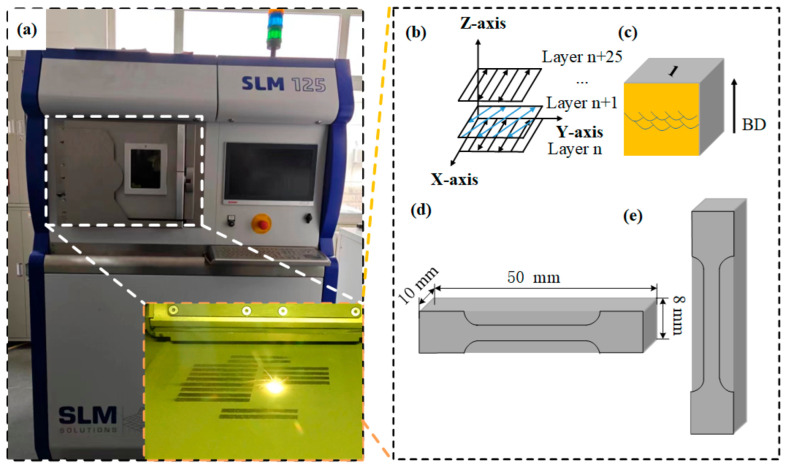
Schematic illustrations (**a**) for selective laser melting (SLM); (**b**) for the scanning strategy with zigzag scanning; (**c**) for the selected vertical section used in relative density calculation, OM, and SEM observation; and (**d**,**e**) for the dimensions of tensile test samples.

**Figure 3 materials-17-02631-f003:**
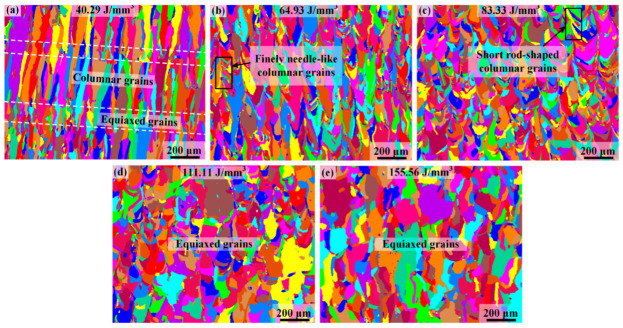
EBSD micrographs showing the grain structures in the SLM-processed Ti-55311 alloy samples produced under different laser energy density conditions: (**a**) 40.29 J/mm^3^, (**b**) 64.93 J/mm^3^, (**c**) 83.33 J/mm^3^, (**d**) 111.11 J/mm^3^, and (**e**) 155.56 J/mm^3^.

**Figure 4 materials-17-02631-f004:**
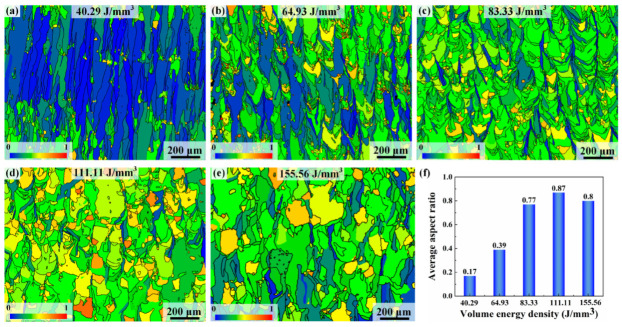
The variation in the aspect ratios of β grains with volume energy density: (**a**) 40.29 J/mm^3^, (**b**) 64.93 J/mm^3^, (**c**) 83.33 J/mm^3^, (**d**) 111.11 J/mm^3^, (**e**) 155.56 J/mm^3^, and (**f**) average aspect ratio.

**Figure 5 materials-17-02631-f005:**
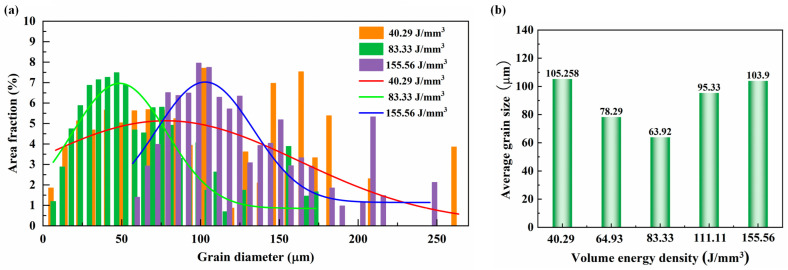
The variation in the grain size of β phase with laser energy density: (**a**) the size distribution at different volume energy densities; (**b**) average grain size.

**Figure 6 materials-17-02631-f006:**
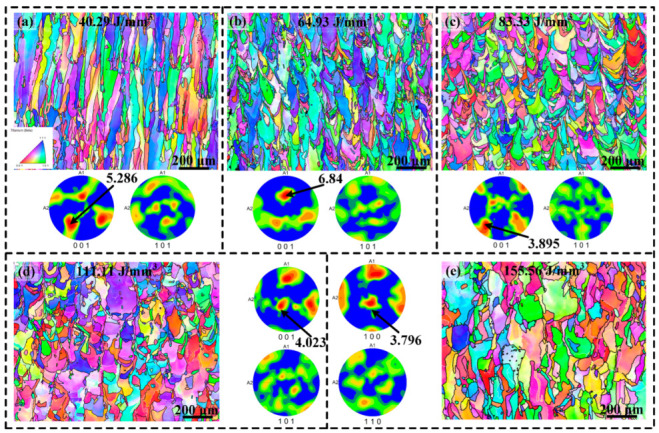
EBSD inverse pole figure (IPF) color maps and pole figures (PF) of the samples made at (**a**) 40.29 J/mm^3^, (**b**) 64.93 J/mm^3^, (**c**) 83.33 J/mm^3^, (**d**) 111.11 J/mm^3^, and (**e**) 155.56 J/mm^3^.

**Figure 7 materials-17-02631-f007:**
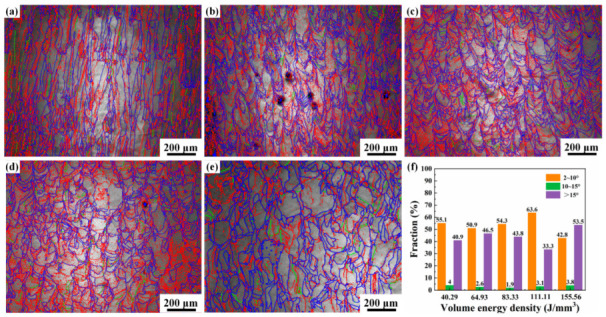
Grain boundary distribution under different volume energy densities: (**a**) 40.29 J/mm^3^, (**b**) 64.93 J/mm^3^, (**c**) 83.33 J/mm^3^, (**d**) 111.11 J/mm^3^, and (**e**) 155.56 J/mm^3^; (**f**) columnar statistical chart, in which (**a**–**e**) red represents low-angle grain boundaries (2~10°), green represents medium-angle grain boundaries (10~15°), and blue represents high-angle grain boundaries (>15°).

**Figure 8 materials-17-02631-f008:**
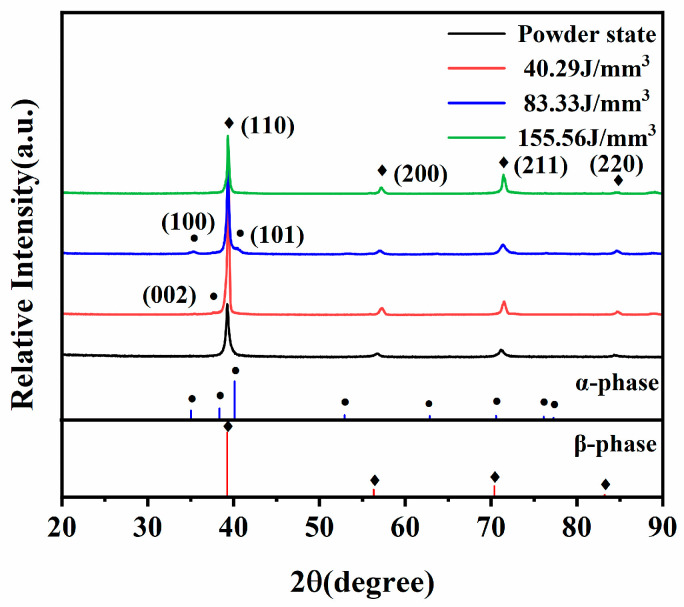
XRD patterns of the SLM-ed samples with different volume energy densities.

**Figure 9 materials-17-02631-f009:**
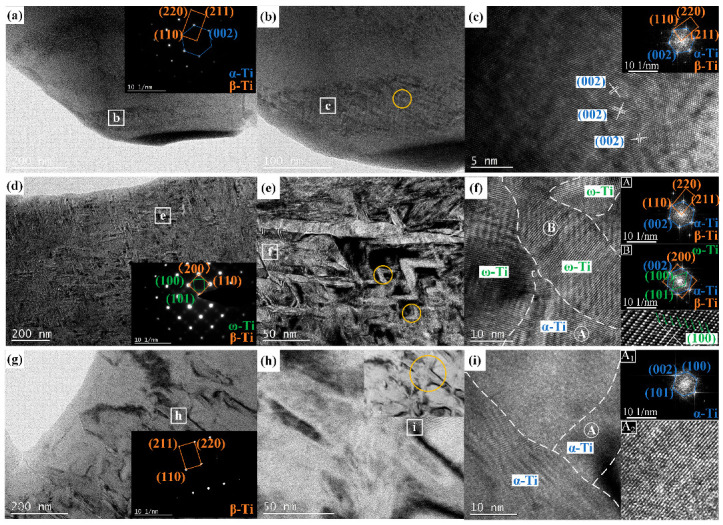
The TEM diagrams of SLM-processed Ti-55311 alloy at different volume energy densities: (**a**–**c**) 40.29 J/mm^3^, (**d**–**f**) 83.33 J/mm^3^, and (**g**–**i**) 155.56 J/mm^3^.

**Figure 10 materials-17-02631-f010:**
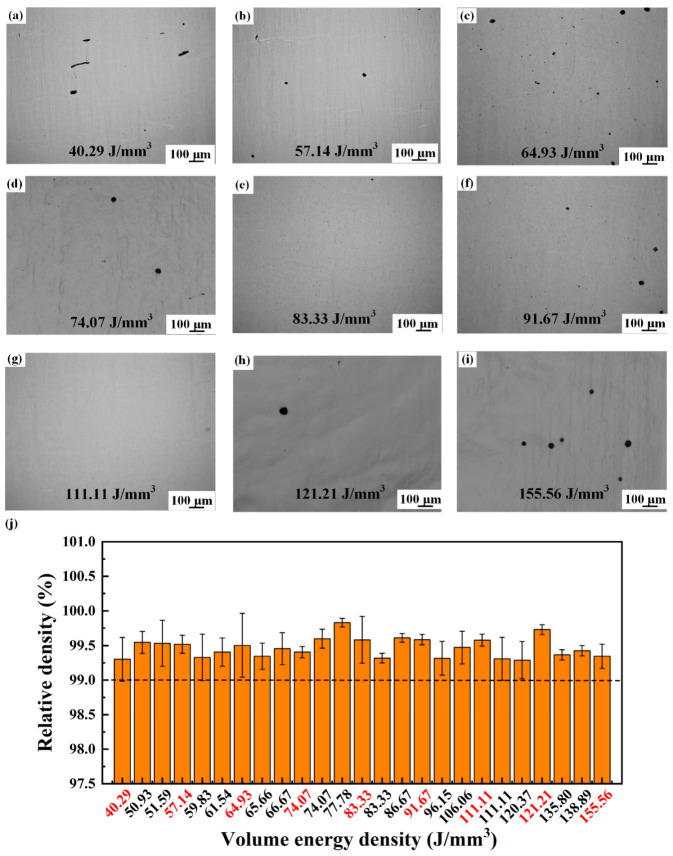
The densification behavior of SLM Ti-55311 alloy at different volume energy densities: (**a**–**i**) OM micrographs of porosity distribution and (**j**) charts of as-printed samples.

**Figure 11 materials-17-02631-f011:**
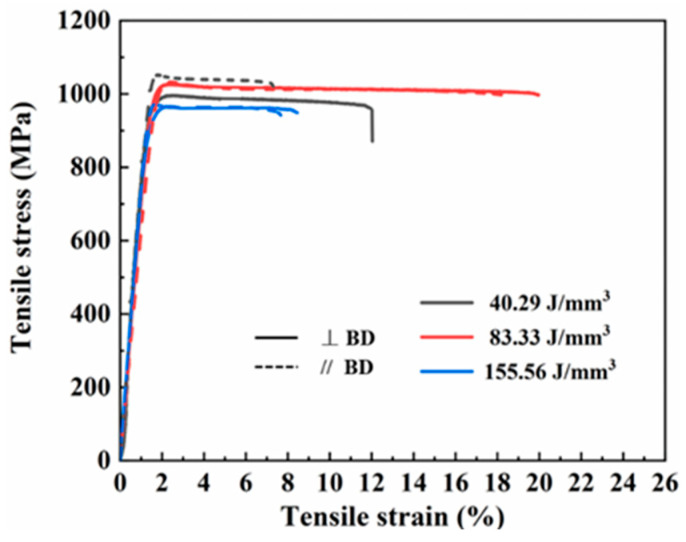
Tensile stress–strain curves along and perpendicular to the building direction under different volume energy densities.

**Figure 12 materials-17-02631-f012:**
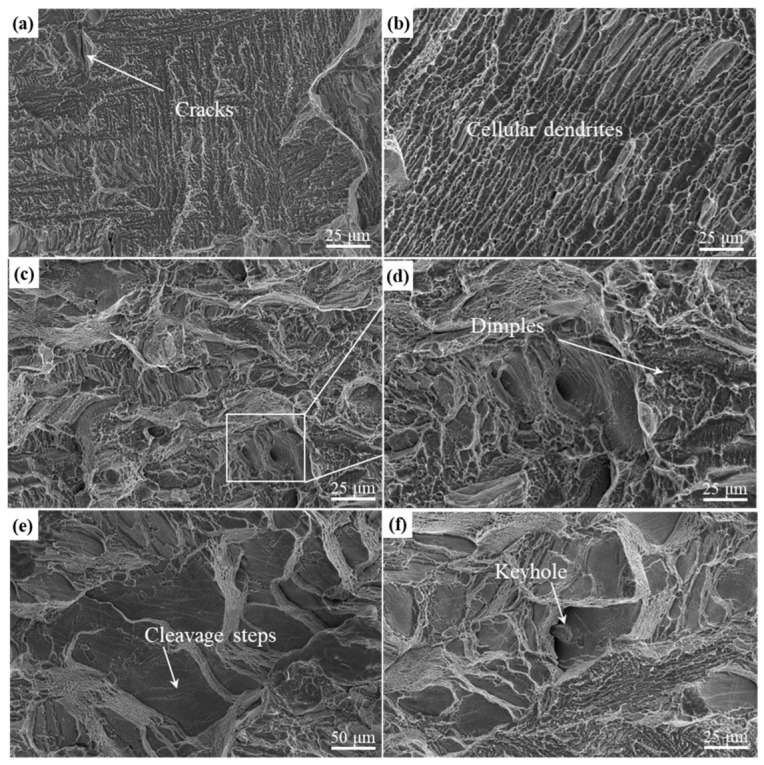
Fracture morphology of Ti-55311 alloy at different volume energy densities: (**a**,**b**) 40.29 J/mm^3^, (**c**,**d**) 83.33 J/mm^3^, and (**e**,**f**) 155.56 J/mm^3^.

**Figure 13 materials-17-02631-f013:**
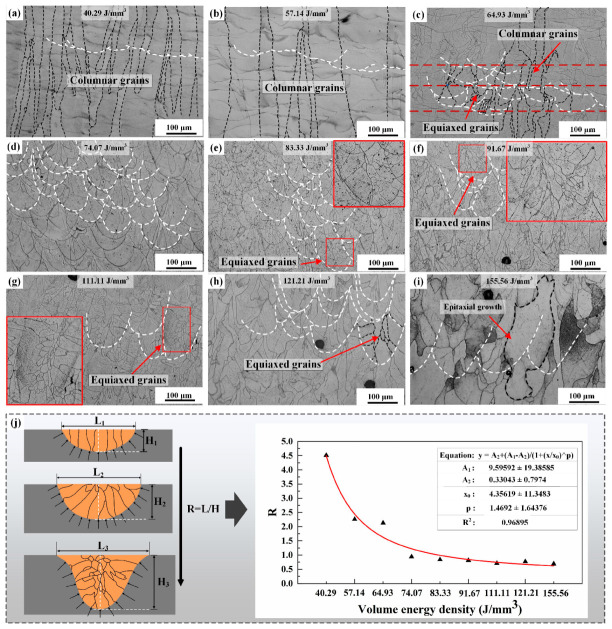
Evolution of melt pool morphology at different volume energy densities, (**a**–**i**) variation in molten pool morphology, (**j**) schematic diagram.

**Figure 14 materials-17-02631-f014:**
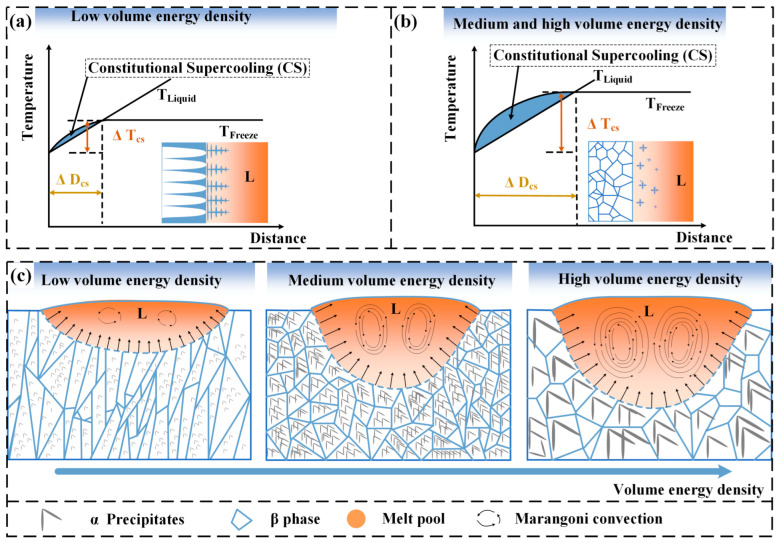
Schematic diagram of the SLM Ti-55311 alloy: (**a**,**b**) constitutive supercooling and (**c**) evolution of grain structure at different volume energy densities.

**Table 1 materials-17-02631-t001:** Chemical composition (wt.%) of Ti-55311 alloy.

Element	Al	Mo	V	Fe	Cr	Ti	C	H	N	O
Percent/wt.%	4.99	5.40	3.07	1.62	1.49	Balance	0.014	0.0023	0.017	0.14

**Table 2 materials-17-02631-t002:** The specific values of strength and plasticity under different conditions.

Volume Energy Density (J/mm^3^)		Yield Strength (MPa)		Elongation (%)	
40.29	//BD	1066.38	1048.9 ± 15.75	4.86	5.45 ± 0.60
		1035.80		6.06	
		1044.52		5.43	
	⊥BD	989.26	988.81 ± 11.65	10.08	10.05 ± 0.58
		1000.23		9.46	
		976.94		10.61	
83.33	//BD	1010.90	1010.53 ± 4.11	16.84	16.59 ± 0.70
		1006.25		17.13	
		1014.44		15.80	
	⊥BD	1003.06	1003.06 ± 2.30	18.43	18.16 ± 0.61
		1016.14		17.46	
		990.15		18.59	
155.56	//BD	970.46	970.46 ± 6.14	6.20	6.1 ± 0.21
		968.09		6.24	
		977.43		5.86	
	⊥BD	948.07	940.13 ± 7.25	6.37	6.8 ± 0.40
		933.86		7.15	
		938.46		6.88	

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
