# Peer review of "Achieving Equiaxed Transition and Excellent Mechanical Properties in a Novel Near-β Titanium Alloy by Regulating the Volume Energy Density of Selective Laser Melting"

_materials, 2024, doi:10.3390/ma17112631_

Round 1

Reviewer 1 Report

Comments and Suggestions for Authors

The authors have submitted a very interesting and well-written manuscript on microstructural transitions in a near-beta titanium alloy driven by volume energy density changes during selective laser melting.

I only had one suggestion as to how to improve the manuscript.  

The mechanical property information presented in Table 2 in the manuscript could be improved by including standard deviations to the yield strength and elongation values given.  Please add the number of samples used per group in the mechanical test to the methods "characterization" section and then add the appropriate standard deviation levels for each group.  Without this information, it is difficult or impossible to determine if the mechanical property results are truly different for different volume energy density groups. 

The mechanical property findings could even be stronger if a simple statistical analysis, such as a one-way ANOVA was performed on the yield strength and elongation mechanical property results.  Please consider this idea as well.

Reviewer 2 Report

Comments and Suggestions for Authors

I reviewed the manuscript entitled “Achieving equiaxed transition and excellent mechanical properties of a novel near-β titanium alloy by regulating the volume energy density of selective laser melting”. The work carried out in the manuscript is interesting However, in my opinion, several aspects should be modified or detailed more in-depth prior to publication, thus major modifications are advised. Here is a list of main comments:

Please consider all the comments below and highlight the changes.

- Conduct a thorough proofreading to correct grammatical errors and ensure smooth syntax throughout the manuscript.

-This work's innovation and importance are not clearly highlighted in the abstract, introduction, and conclusions. Please work on this and prove to us why this work is valuable.

- The abstract is well-written and provides a clear overview of the study. However, consider expanding on the significance of achieving equiaxed grain structures in titanium alloys in the abstract.

-In the introduction, provide a concise overview of the SLM process before delving into specific challenges and strategies. Also, Is there a need to highlight any recent advancements or trends in the field of SLM for titanium alloys?

- Include more recent studies (if available) to ensure the review encompasses the latest developments in the field.

-Can the review critically analyze the limitations and challenges of existing approaches to controlling grain structures?

- Explain the significance of the selected volume energy density values and their relevance to real-world applications.

- Explain why the alloy maintains high relative density despite variations in volume energy density.

- Discuss the potential reasons behind the observed reduction in strength and plasticity at higher volume energy densities.

- In the conclusion section: provide a summary of the key findings regarding the effect of volume energy density on microstructure and mechanical properties.

-Are there any recommendations for future research or potential industrial applications based on the findings of this study?

Comments on the Quality of English Language

Conduct a thorough proofreading to correct grammatical errors and ensure smooth syntax throughout the manuscript.

Round 2

Reviewer 2 Report

Comments and Suggestions for Authors

The requested modifications have been completed as specified.